# SoCal: Selective Oracle Questioning for Consistency-based Active Learning of Cardiac Signals

## Abstract

The ubiquity and rate of collection of cardiac signals produce large, unlabelled datasets. Active learning (AL) can exploit such datasets by incorporating human annotators (oracles) to improve generalization performance. However, the over-reliance of existing algorithms on oracles continues to burden physicians. To minimize this burden, we propose SoCal, a consistency-based AL framework that dynamically determines whether to request a label from an oracle or to generate a pseudo-label instead. We show that our framework decreases the labelling burden while maintaining strong performance, even in the presence of a noisy oracle.

## 1 Introduction

The success of modern-day deep learning algorithms in the medical domain has been contingent upon the availability of large, labelled datasets (Poplin et al., 2018; Tomašev et al., 2019; Attia et al., 2019). Curating such datasets, however, is a challenge due to the time-consuming nature of, and high costs associated with, labelling. This is particularly the case in the medical domain where the input of expert medical professionals is required. One way of overcoming this challenge and exploiting large, *unlabelled* datasets is via the active learning (AL) framework (Settles, 2009). This framework iterates over three main steps: 1) a learner is tasked with acquiring unlabelled instances, usually through an acquisition function, 2) an oracle (e.g. physician) is tasked with labelling such acquired instances, and 3) the learner is trained on the existing and newly-labelled instances.

By altering the way in which acquisitions are performed and the degree of involvement of the oracle, the active learning framework aims to improve the performance of a network while minimizing the burden of labelling on the oracle. One principal desideratum for an acquisition function is its ability to reduce the size of the version space, the set of hypotheses (decision boundaries) consistent with the labelled training instances. This ability is highly dependent upon the approximation of the version space, a goal that Monte Carlo Dropout (MCD) attempts to achieve (see Fig. 1a). For example, state-of-the-art uncertainty-based acquisition functions, such as BALD (Houlsby et al., 2011), used alongside MCD acquire instances that lie in a region of uncertainty, a region where there is high disagreement between the hypotheses about a particular instance. In many scenarios, however, estimating this region of uncertainty is nontrivial. Furthermore, existing AL frameworks are overly *reliant* on the presence of an oracle. Such over-reliance precludes the applicability of AL algorithms to certain environments, such as low-resource healthcare settings, where an oracle is either unavailable or ill-trained for the task at hand.

**Contributions.** In this work, we aim to design an active learning framework that better estimates the region of uncertainty and decreases its reliance on an oracle. Our contributions are as follows:

1. **Consistency-based active learning framework**: we propose a novel framework that stochastically perturbs inputs, network parameters, or both to guide the acquisition of unlabelled instances.

2. **Selective oracle questioning**: we propose a dynamic strategy which learns, for an acquired unlabelled instance, whether to request a label from an oracle or to generate a pseudo-label instead.

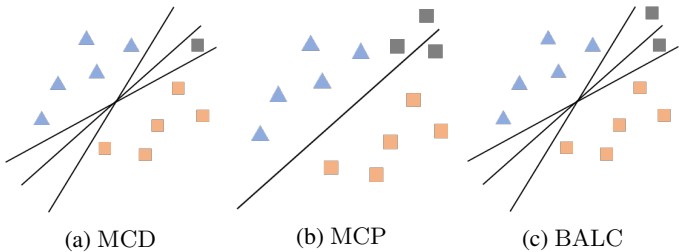

(a) MCD  (b) MCP  (c) BALC

Figure 1: Labelled instances from two classes and unlabelled instances (gray) alongside the Version space of (a) MCD where each MC sample is viewed as a distinct hypothesis (decision boundary), (b) MCP where there is one hypothesis but several perturbations of the unlabelled instance, and (c) BALC where there are several hypotheses in addition to the unlabelled instance and its perturbed counterpart.

## 2 Related Work

**Active learning** methdologies were recently reviewed by Settles (2009). In the healthcare domain, Gong et al. (2019) propose to acquire instances from an electronic health record (EHR) database using a Bayesian deep latent Gaussian model to improve mortality prediction. Smailagic et al. (2018; 2019) acquire unannotated medical images by measuring their distance in a latent space to images in the training set. The work of Wang et al. (2019) is similar to ours in that they focus on the electrocardiogram (ECG). Gal et al. (2017) adopt BALD (Houlsby et al., 2011) in the context of Monte Carlo Dropout to acquire datapoints that maximize the Jensen-Shannon divergence (JSD) across MC samples. Previous work attempts to learn from multiple or imperfect oracles (Dekel et al., 2012; Zhang & Chaudhuri, 2015; Sinha et al., 2019). For example, Urner et al. (2012) propose choosing the oracle that should label a particular instance. Unlike our approach, they do not explore independence from an oracle. Yan et al. (2016) consider oracle abstention in an AL setting. Instead, we place the decision of abstention under the control of the learner. To the best of our knowledge, previous work, in contrast to ours, has assumed the existence of an oracle and has not explored a dynamic oracle selection strategy.

**Consistency training** in the context of semi-supervised learning helps enforce the smoothness assumption (Zhu, 2005). For example, Interpolation Consistency Training (Verma et al., 2019) penalizes networks for not generating a linear combination of outputs in response to a linear combination of inputs. Similarly, Xie et al. (2019) penalizes networks for generating drastically different outputs in response to perturbed instances. In the process, networks learn perturbation-invariant representations. McCallumzy & Nigamy (1998) introduce an acquisition function that calculates the average Kullback-Leibler divergence, $\mathcal{D}_{KL}$, between the output of a network and the consensus output across all networks in an ensemble. Unlike ours, their approach does not exploit perturbations. Similar to our work is that of Gao et al. (2019) which incorporates into the objective function a consistency-loss based on the $\mathcal{D}_{KL}$ and actively acquires instances using the variance of the probability assigned to each class by the network in response to perturbed versions of the same instance.

**Selective classification** imbues a network with the ability to abstain from making predictions. Chow (1970); El-Yaniv & Wiener (2010) introduce the risk-coverage trade-off whereby the empirical risk of a model is inversely related to its rate of abstentions. Wiener & El-Yaniv (2011) use a support vector machine (SVM) to rank and reject instances based on the degree of disagreement between hypotheses. In some frameworks, these are the same instances that active learning views as most informative. Cortes et al. (2016) outline an objective function that penalizes abstentions that are inappropriate and frequent. Most recently, Liu et al. (2019) propose the gambler's loss to learn a selection function that determines whether instances are rejected. However, this approach is not implemented in the context of AL. Most similar to our work is SelectiveNet (Geifman & El-Yaniv, 2019) where a multi-head architecture is used alongside an empirical selective risk objective function and a percentile threshold. However, their work assumes the presence of ground-truth labels and, therefore, does not extend to unlabelled instances.

## 3    BACKGROUND

### 3.1    ACTIVE LEARNING

Consider a learner $f_\omega : x \in \mathbb{R}^m \to v \in \mathbb{R}^d$, parameterized by $\omega$, that maps an $m$-dimensional input, $x$, to a $d$-dimensional representation, $v$. Further consider $g_\phi : v \in \mathbb{R}^d \to y \in \mathbb{R}^C$ that maps a $d$-dimensional representation, $v$, to a $C$-dimensional output, $y$, where $C$ is the number of classes. After training on a pool of labelled data $L = (X_L, Y_L)$ for $\tau$ epochs, the learner is tasked with querying the unlabelled pool of data $U = (X_U, Y_U)$ and acquiring the top $b$ fraction of instances, $x_b \sim X_U$, that it deems to be most informative. The degree of informativeness of an instance is determined by an acquisition function, $\alpha$, such as BALD (Houlsby et al., 2011). Additional acquisition functions can be found in Appendix A. These are typically used in conjunction with Monte Carlo Dropout (Gal & Ghahramani, 2016) to identify instances that lie in the region of uncertainty, a region in which hypotheses disagree the most about instances.

## 4    METHODS

### 4.1    CONSISTENCY-BASED ACTIVE LEARNING

#### 4.1.1    MONTE CARLO PERTURBATIONS

Unlabelled instances in proximity to the decision boundary are likely to be more informative for training than those further away. To identify such instances, we stochastically perturb them and observe the network's outputs. The intuition is that such outputs will differ significantly across the perturbations for instances close to the decision boundary (see Fig. 1b). We refer to this setup as Monte Carlo Perturbations (MCP) and illustrate its derivation in Appendix B.

#### 4.1.2    BAYESIAN ACTIVE LEARNING BY CONSISTENCY

Acquisition functions dependent upon perturbations applied to either the inputs (MCP) or the network parameters (MCD) alone can fail to identify instances that lie in the region of uncertainty. We illustrate this point with the following example: without loss of generality, let us assume an unlabelled instance is in proximity to some decision boundary A and is classified by the network as belonging to some arbitrary class 3. Such proximity should deem the instance informative for the training process (Settles, 2009). In the MCD setting, perturbations are applied to parameters, generating various decision boundaries, which in turn influence the network outputs. In Fig. 2 (red rectangle), we visualize such outputs for three arbitrary classes. If these parameter perturbations happen to be too small in magnitude, for example, then the network will continue to classify the instance as belonging to the same class. At this stage, regardless of whether an uncertainty-based or a consistency-based acquisition function is used, the instance would be deemed *uninformative*, and thus not acquired. As a result, an instance that should have been acquired (due to its proximity to the decision boundary) was *erroneously* deemed uninformative. A similar argument can be extended to MCP.

By applying perturbations to both instances and network parameters, we aim to leverage the smoothness assumption (Zhu, 2005) to better identify instances that lie in the region of uncertainty and thus avoid missing their acquisition. Motivated by this, we propose a framework, entitled Bayesian Active Learning by Consistency (BALC) (see Fig. 1c), that consists of three main steps: 1) we perturb an instance, $x$, to generate $z$, 2) we perturb the network parameters, $\omega$, to generate $\omega'$, and 3) we pass both instances through the *perturbed* network, generating outputs, $p(y|x, \omega')$ and $p(y|z, \omega') \in \mathbb{R}^C$, respectively. We perform these steps for $T$ stochastic perturbations and generate two matrices of network outputs, $G(x), G'(z) \in \mathbb{R}^{T \times C}$. We visualize such network outputs in Fig. 2 where $T = 3$ and $C = 3$.

To leverage $G$ and $G'$, we propose two divergence-based acquisition functions that acquire instances that the network is least robust to. In $\text{BALC}_{\text{KLD}}$, we calculate the $D_{KL}$ between two $C$-dimensional Gaussians that are empirically fit to $G$ and $G'$.

$$\text{BALC}_{\text{KLD}} = \mathcal{D}_{KL}(\mathcal{N}(\mu(x), \Sigma(x) \parallel \mathcal{N}(\mu(z), \Sigma(z)))) \tag{1}$$

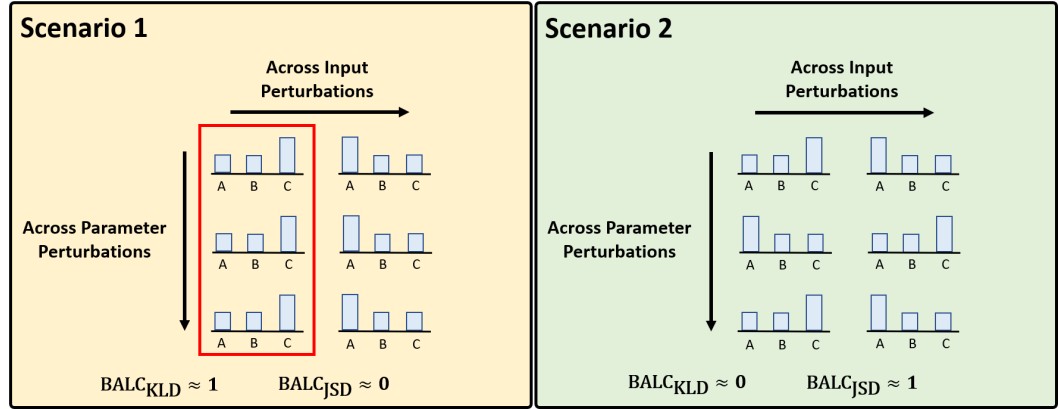

Figure 2: **(Scenario 1)** network output variations (for three classes, A, B, and C) caused primarily by input perturbations. The red rectangle illustrates a potential limitation of MCD. Given that these parameter perturbations do not result in network output variations and MCD is dependent upon said perturbations, unlabelled instances can be erroneously deemed uninformative. **(Scenario 2)** network output variations caused by both input and parameter perturbations. We show that while $BALC_{KLD}$ is likely to acquire instances due to input perturbations, $BALC_{JSD}$ considers both input and parameter perturbations when performing acquisitions.

where $\mu = \frac{1}{T}\sum_i^T G$ and $\Sigma = (G - \mu)^T(G - \mu)$ represent the empirical mean vector and covariance matrix of the network outputs, respectively. $BALC_{KLD}$ is likely to detect output variations due to input perturbations. We support this claim in Fig. 2 by illustrating two scenarios. In scenario 1, network output variations are caused solely by input perturbations. In contrast, in scenario 2, network output variations are caused by both input and parameter perturbations. We show that $BALC_{KLD} \approx 1$ and 0 in these two scenarios, respectively. Since the higher the value of an acquisition function, the more informative an instance is, these scenarios illustrate $BALC_{KLD}$'s preference for input perturbations. To detect variations due to both input *and* parameter perturbations, we introduce $BALC_{JSD}$ whose full derivation can be found in Appendix C.

$$BALC_{JSD} = \overbrace{\mathbb{E}_{i \in T}\left[\mathcal{D}_{KL}(G_i(x) \parallel G'_i(z))\right]}^{\text{across parameter perturbations}} - \overbrace{\mathcal{D}_{KL}(\mathbb{E}_{i \in T}[G(x)] \parallel \mathbb{E}_{i \in T}[G'(z)])}^{\text{across input perturbations}} \qquad (2)$$

## 4.2 TRACKED ACQUISITION FUNCTION

Deriving the informativeness of unlabelled instances based solely on the value of the acquisition function at a *single* epoch can be erroneous. This is partially driven by limitations in the approximation of the version space, which is known to hinder performance (Cohn et al., 1994). To improve this approximation, we propose to *track* an acquisition function over time (e.g., epochs) before employing it to acquire instances. The intuition is that by incorporating temporal information, we accumulate hypotheses in the version space and thus obtain a more reliable estimate of the relative informativeness of each instance. Acquiring such instances would help reduce the size of the version space at a greater rate. For any tracked acquisition function, $\alpha(t)$, the corresponding area under the temporal acquisition function, $AUTAF \in \mathbb{R}^1$, is calculated as follows:

$$AUTAF = \int_0^\tau \alpha(t)dt \approx \sum_{t=0}^\tau \left(\frac{\alpha(t + \Delta t) + \alpha(t)}{2}\right)\Delta t \qquad (3)$$

where the integral is approximated using the trapezoidal rule, $\Delta t$ is the time-step (in epochs) between epochs at which ordinary acquisition values are calculated, and $\tau$ is the epoch at which the AUTAF is calculated and an acquisition of unlabelled instances is performed.

## 4.3 SELECTIVE ORACLE QUESTIONING

Alongside our consistency-based AL framework, we aim to minimize the burden of labelling on an oracle. To do so, we learn a network that dynamically determines whether to request a label from an

oracle or to generate a pseudo-label for each acquired unlabelled instance. We refer to this strategy as selective oracle questioning in active learning (SoQal).

**Oracle selection network.** In addition to the learners outlined in Sec. 3.1, $f_\omega$ and $g_\phi$, we introduce an oracle selection network, $h_\theta : v \in \mathbb{R}^d \to t \in [0,1]$ parameterized by $\theta$, that maps $d$-dimensional representations, $v$, to a scalar, $t$, as shown in Fig. 3.

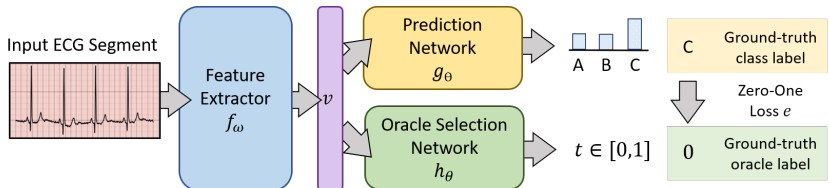

Figure 3: An instance, $x$, is provided as input to the feature extractor, $f_\omega$. The representation, $v$, is passed through $g_\phi$ to generate task predictions, and through $h_\theta$ to help determine whether a label is requested from an oracle. The zero-one loss of $g_\phi$ acts as the ground-truth label for $t$ where $t \approx 1$ indicates a hard-to-classify instance that would benefit from an oracle label. Otherwise, the instance is pseudo-labelled by taking the argmax of the outputs of the prediction network.

**Proxy for misclassifications.** Ideally, a network should only be reliant on an oracle when it misclassifies an instance itself. Quantifying these misclassifications is trivial in the presence of ground-truth labels. However, in the AL framework, we are interested in making decisions on *unlabelled* instances. Therefore, we need a reliable proxy for such misclassifications. We propose that this proxy be the output of the oracle selection network, $t$. For example, low and high values of $t$ can indicate correct and incorrect network predictions, respectively. To learn this behaviour, $h_\theta$ needs an appropriate supervisory signal. For this, we choose the zero-one loss, $e$, of the prediction network, $g_\phi$, as the ground-truth label (see Fig. 3). For a mini-batch of size, $B$, our objective function thus consists of two terms: 1) a cross-entropy class prediction loss for the main task, and 2) a binary cross-entropy oracle selection loss, with a weighting coefficient, $\beta$, (described next).

$$\mathcal{L} = \sum_{i=1}^{B} \overbrace{-\log\left(p(y_i = c|x_i, \omega, \phi)\right)}^{\text{class prediction loss}} \overbrace{-\beta e_i \log\left(h_\theta(t|x_i)\right) - (1 - e_i)\log\left(1 - h_\theta(t|x_i)\right)}^{\text{oracle selection loss}} \quad (4)$$

where $c$ is the target class. During the early stages of training on *labelled* data, a network struggles to classify instances correctly. This means that the ratio of zero to one losses will be skewed toward the latter. As training progresses and the network becomes more adept at classifying instances, this ratio becomes skewed in the opposite direction. Such an imbalance in the ground-truth labels, $e$, and their subsequent shift send strong supervisory signals to $h_\theta$. For example, during the early stages of training, the outputs of $h_\theta$ will be high and pulled towards $t = 1$ (high error) even if the corresponding instance was correctly classified. Such behaviour makes it difficult to ascertain whether instances have been misclassified, hindering the reliability of $t$ as a proxy. To offset the aforementioned class imbalance, we introduce a dynamic hyperparameter, $\beta = \frac{\sum \delta_{e=0}}{\sum \delta_{e=1}}$, where $\delta$ is the Kronecker delta function. As training progress, $\beta < 1 \to \beta > 1$, as the ratio of correctly classified ($e = 0$) to misclassified ($e = 1$) instances within a mini-batch changes.

**Decision-making with proxy.** We aim to exploit $t$ for the binary decision of either requesting a label from an oracle or generating a pseudo-label. One way to do so is via a simple threshold at $0.5$. However, this threshold may not be appropriate. In designing a robust selection strategy, we must account for the distribution of the $t$ values that corresponds to each decision and the separability of such distributions. In Figs. 4a and 4b, we illustrate these distributions during the early and late stages of training, colour-coded based on whether the $t$ values correspond to correctly-classified ($e = 0$) or misclassified ($e = 1$) *labelled* training instances.

We now outline how the binary decision is made. After each training epoch, we fit the $t$ values in Fig. 4b to two unimodal Gaussian distributions. This generates $\mathcal{N}_0(\mu_0, \sigma_0^2)$ and $\mathcal{N}_1(\mu_1, \sigma_1^2)$ for $e = 0$ and $e = 1$, respectively. We choose to quantify the separability of these two distributions using the Hellinger distance, $\mathcal{D}_H \in [0, 1]$, as it allows for a straightforward threshold. Low separability expressed as $\mathcal{D}_H < S$ implies that $h_\theta$ has yet to generate a reliable proxy and thus an oracle is

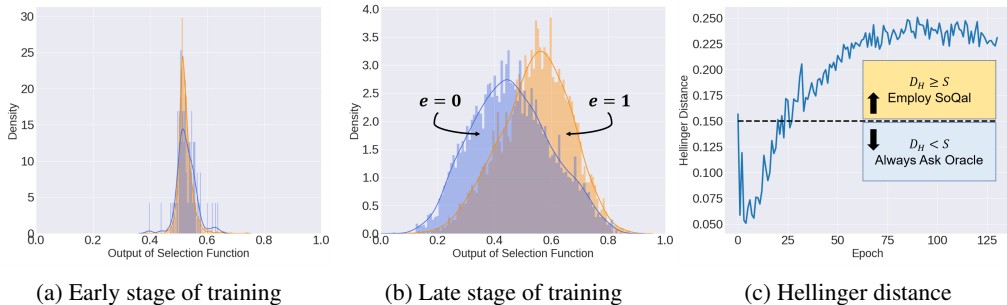

Figure 4: Density of the outputs, $t$, of $g_\theta$ colour-coded based on the zero-one classification error during the (a) early and (b) late stages of training. (c) the Hellinger distance, $\mathcal{D}_H$, between the two distributions of $t$ during training increases as they become more separable.

requested for a label. We note that the value of $S$ can be altered depending on the relative level of trust one has in the network and oracle. When $\mathcal{D}_H \geq S$, we evaluate $\mathcal{N}_0$ and $\mathcal{N}_1$ at the $t$ output for each acquired unlabelled instance and define $p(A)$ as the probability of requesting a label from an oracle. We elucidate the entire active learning algorithm in Appendix E.

$$p(A) = \begin{cases} 1, & \mathcal{D}_H < S \\ 1, & \mathcal{N}\left(t|\mu_1, \sigma_1^2, e=1\right) > \mathcal{N}\left(t|\mu_0, \sigma_0^2, e=0\right) \text{ and } \mathcal{D}_H \geq S \\ 0, & \text{otherwise} \end{cases} \tag{5}$$

## 5 EXPERIMENTAL DESIGN

### 5.1 DATASETS

We conduct experiments in PyTorch (Paszke et al., 2019) on four publically-available datasets. These datasets consist of cardiac time-series data such as the photoplethysmogram (PPG) and the electrocardiogram (ECG) alongside cardiac arrhythmia labels. We use $\mathcal{D}_1$ = PhysioNet 2015 PPG, $\mathcal{D}_2$ = PhysioNet 2015 ECG (Clifford et al., 2015) (5-way), $\mathcal{D}_3$ = PhysioNet 2017 ECG (Clifford et al., 2017) (4-way), $\mathcal{D}_4$ = Cardiology ECG (Hannun et al., 2019) (12-way), and $\mathcal{D}_5$ = CIFAR10 Krizhevsky et al. (2009) (10-way).

To observe the impact of the availability of labelled training data on the active learning procedure, we take a fraction $\beta = (0.1, 0.3, 0.5, 0.7, 0.9)$ of the training dataset and place it into the labelled set. Its complement is placed into the unlabelled set. Details about the data splits, preprocessing steps, and the network architecture can be found in Appendices F and G.

### 5.2 BASELINES

**Acquisition Functions.** We compare our novel active learning framework to the state-of-the-art acquisition functions used in conjunction with MCD. These include **Var Ratio**, **Entropy**, and **BALD**, definitions of which can be found in Appendix A. We also compare to the scenario in which active learning is not employed (**No AL**).

**Selective Oracle Questioning.** We experiment with baselines that exhibit varying degrees of oracle dependence. **No Oracle** is a scenario in which 0% of the labels that correspond to unlabelled instances are oracle-based and are instead pseudo-labelled by taking the argmax of the network predictions. **Epsilon Greedy** is a stochastic strategy Watkins (1989) that we adapt to exponentially decay the reliance of the network on an oracle as a function of the number of acquisition epochs. **Entropy Response** assumes that high entropy predictions generated by a network are indicative of instances that the network is unsure of. Therefore, we introduce a threshold, $S_{\text{Entropy}}$, such that if it is exceeded, an oracle is requested to label the chosen instance (see Appendix G). The most dependent baseline is **100% Oracle**, a traditionally-employed strategy in AL where 100% of the labels are oracle-based.

We do not compare our methods to Softmax Response (Geifman & El-Yaniv, 2017) and SelectiveNet (Geifman & El-Yaniv, 2019), despite their strong performance for selective classification, as they do not trivially extend to the setting in which labels are unavailable.

### 5.3 Hyperparameters

For all experiments, we chose the number of MC samples $T = 20$ to balance between computational complexity and accuracy of the approximation of the version space. We acquire unlabelled instances at pre-defined epochs during training which we refer to as acquisition epochs, $\tau = 5n$, $n \in \mathbb{N}^+$. During each acquisition epoch, we acquire $b = 2\%$ of the remaining unlabelled instances. We also investigate the effect of such hyperparameters on performance (see Appendices P-R). When experimenting with tracked acquisition functions, we chose the temporal period, $\Delta t = 1$. For the CIFAR10 experiments, we chose $T = 5$, $\tau = 2n$, and $b = 10\%$.

**Selective Oracle Questioning.** Recall that we delegate selective oracle questioning to the network only when $\mathcal{D}_H \geq S$. Given $\mathcal{D}_H$'s increasing trend during training (see Fig. 4c), we chose $S = 0.15$ to balance between the reliability of the proxy and the independence of the network from an oracle. We also explore the sensitivity of SoQal to this choice of $S$.

## 6 Experimental Results

### 6.1 Active Learning without Oracle

Active learning frameworks typically assume the presence of an oracle (expert annotator). However, oracles are not always available, particularly in the medical domain where physicians have limited time to provide annotations. To reflect this scenario, we evaluate the ability of our AL framework to operate *without* an oracle. In Fig. 5a, we illustrate the validation AUC of various methods when exposed to a fraction, $\beta$, of the labelled training data on $\mathcal{D}_2$ and $\mathcal{D}_5$, respectively.

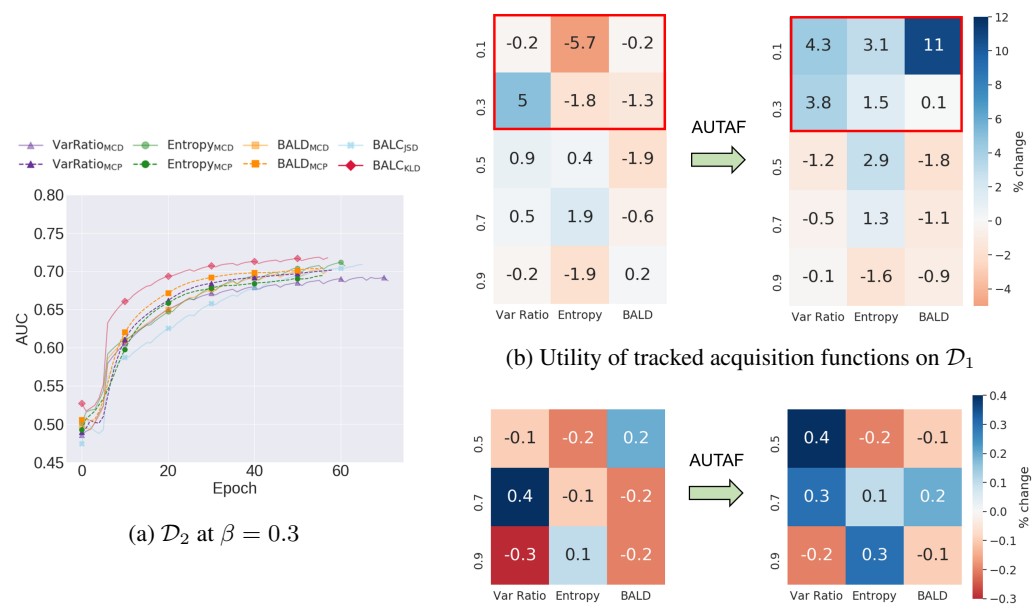

(a) $\mathcal{D}_2$ at $\beta = 0.3$

(b) Utility of tracked acquisition functions on $\mathcal{D}_1$

(c) Utility of tracked acquisition functions on $\mathcal{D}_5$

Figure 5: Validation AUC on **(a)** $\mathcal{D}_2$ at $\beta = 0.3$. Mean percent change in test AUC when comparing MCP with static and tracked acquisition functions to MCD with their static counterparts on **(b)** $\mathcal{D}_1$ and **(c)** $\mathcal{D}_5$. We show results for Var Ratio, Entropy, and BALD, at all fractions, $\beta \in [0.1, 0.3, 0.5, 0.7, 0.9]$ and across five seeds.

We find that $BALC_{KLD}$ achieves strong generalization performance and does so in a fewer number of epochs relative to the remaining methods. For example, in Fig. 5a, $BALC_{KLD}$ achieves an AUC $\approx$ 0.69 after only 20 epochs whereas $BALD_{MCD}$ does so at epoch 40. This implies that $BALC_{KLD}$ can result in a two-fold reduction in training time. It also achieves a higher final AUC $\approx$ 0.72 relative to the remaining methods. We hypothesize that such behaviour is due to $BALC_{KLD}$'s improved ability to estimate the region of uncertainty, and thus acquires more informative instances. Moreover, given the absence of an oracle, these informative instances are likely to have also been pseudo-labelled correctly. The acquisition of more informative instances which are labelled correctly by the network suggests that such instances are *closer* to the decision boundary than their non-acquired counterparts yet are still on the correct side of the boundary. We arrive at similar conclusions for the remaining experiments (see Appendix H and J).

Having illustrated the potential benefit of static acquisition functions, we now move on to quantify the effect of incorporating temporal information on generalization performance. In Figs. 5b and 5c, we illustrate the percent change in the AUC when using MCP, with and without tracked acquisition functions, relative to MCD. We find that tracked acquisition functions are most useful when the initial size of the labelled dataset is small ($\downarrow \beta$ values) (red rectangle). For example, incorporating temporal information into BALD at $\beta = 0.1$ improves the generalization performance by $11\%$. We hypothesize that this improvement is due to the increased enumeration of hypotheses over time, which in turn, results in a more reliable approximation of the version space.

## 6.2 ACTIVE LEARNING WITH NOISE-FREE ORACLE

In this section, we relax the assumption that physicians are unavailable for annotation purposes. Instead, we focus on alleviating the labelling burden that is placed on physicians when they are available. More specifically, we assume that oracles can provide accurate labels, i.e., noise-free. In Table 1, we illustrate the test-set AUC of the oracle questioning strategies on all datasets at $\beta = 0.1$.

We find that SoQal consistently outperforms its counterparts across $\mathcal{D}_1$ - $\mathcal{D}_3$. For example, while using $BALD_{MCD}$ on $\mathcal{D}_2$, SoQal achieves an AUC = 0.707 whereas Epsilon Greedy and Entropy Response achieve AUC = 0.609 and 0.584, respectively. Such a finding suggests that SoQal is better equipped to know *when* and for which *instance* a label should be requested from an oracle. One could argue that SoQal's superiority is due to its high dependence on the oracle. In fact, we show that this is not the case in Appendix L. On the other hand, we observe that SoQal performs on par with the other methods on $\mathcal{D}_4$ and $\mathcal{D}_5$. We hypothesize that this outcome is due to the cold-start problem (Konyushkova et al., 2017) where AL algorithms fail to learn due to the limited availability of labelled training data. We support this claim with experiments in Appendix M. Moreover, we remind readers that by increasing the value of $S$ in the SoQal experiments, networks can cede more control to the oracle and thus further improve performance, an effect we quantify in Appendix N.

Table 1: Mean test AUC of oracle questioning strategies in the presence of a noise-free oracle. Results are shown for a subset of the acquisition functions on $\mathcal{D}_1 - \mathcal{D}_5$ and are averaged across five seeds.

| Dataset | Ac. Function $\alpha$ | Oracle Questioning Method | | | | | No AL |
|---|---|---|---|---|---|---|---|
| | | No Oracle | Entropy Response | Epsilon Greedy | SoQal (ours) | 100% Oracle | |
| $\mathcal{D}_1$ | $BALD_{MCD}$ | $0.465 \pm 0.017$ | $0.496 \pm 0.039$ | $0.491 \pm 0.028$ | $\mathbf{0.621 \pm 0.021}$ | $0.653 \pm 0.013$ | $0.577 \pm 0.014$ |
| | $BALD_{MCP}$ | $0.464 \pm 0.023$ | $0.517 \pm 0.043$ | $0.501 \pm 0.043$ | $\mathbf{0.645 \pm 0.015}$ | $0.676 \pm 0.020$ | |
| | $BALC_{KLD}$ | $0.500 \pm 0.023$ | $0.548 \pm 0.034$ | $0.548 \pm 0.042$ | $\mathbf{0.598 \pm 0.055}$ | $0.634 \pm 0.030$ | |
| | Temporal $BALC_{KLD}$ | $0.496 \pm 0.024$ | $0.536 \pm 0.040$ | $0.521 \pm 0.059$ | $\mathbf{0.646 \pm 0.067}$ | $0.659 \pm 0.033$ | |
| $\mathcal{D}_2$ | $BALD_{MCD}$ | $0.573 \pm 0.063$ | $0.584 \pm 0.041$ | $0.609 \pm 0.071$ | $\mathbf{0.707 \pm 0.038}$ | $0.713 \pm 0.053$ | $0.679 \pm 0.040$ |
| | $BALD_{MCP}$ | $0.589 \pm 0.045$ | $0.638 \pm 0.043$ | $0.637 \pm 0.044$ | $\mathbf{0.677 \pm 0.042}$ | $0.735 \pm 0.028$ | |
| | $BALC_{KLD}$ | $0.602 \pm 0.044$ | $0.582 \pm 0.017$ | $0.643 \pm 0.033$ | $\mathbf{0.677 \pm 0.024}$ | $0.722 \pm 0.018$ | |
| | Temporal $BALC_{KLD}$ | $0.575 \pm 0.017$ | $0.612 \pm 0.018$ | $0.605 \pm 0.019$ | $\mathbf{0.648 \pm 0.057}$ | $0.735 \pm 0.011$ | |
| $\mathcal{D}_3$ | $BALD_{MCD}$ | $0.581 \pm 0.014$ | $0.588 \pm 0.013$ | $0.673 \pm 0.015$ | $\mathbf{0.721 \pm 0.025}$ | $0.802 \pm 0.008$ | $0.716 \pm 0.012$ |
| | $BALD_{MCP}$ | $0.623 \pm 0.020$ | $0.676 \pm 0.058$ | $0.665 \pm 0.028$ | $\mathbf{0.720 \pm 0.044}$ | $0.798 \pm 0.007$ | |
| | $BALC_{KLD}$ | $0.631 \pm 0.010$ | $0.629 \pm 0.004$ | $0.643 \pm 0.041$ | $\mathbf{0.731 \pm 0.033}$ | $0.787 \pm 0.008$ | |
| | Temporal $BALC_{KLD}$ | $0.600 \pm 0.005$ | $0.654 \pm 0.019$ | $0.654 \pm 0.019$ | $\mathbf{0.730 \pm 0.024}$ | $0.794 \pm 0.002$ | |
| $\mathcal{D}_4$ | $BALD_{MCD}$ | $0.486 \pm 0.011$ | $0.489 \pm 0.030$ | $0.474 \pm 0.037$ | $0.468 \pm 0.021$ | $0.585 \pm 0.011$ | $0.486 \pm 0.023$ |
| | $BALD_{MCP}$ | $0.493 \pm 0.030$ | $0.504 \pm 0.026$ | $0.492 \pm 0.024$ | $0.499 \pm 0.029$ | $0.605 \pm 0.024$ | |
| | $BALC_{KLD}$ | $0.505 \pm 0.032$ | $0.504 \pm 0.039$ | $0.473 \pm 0.010$ | $0.495 \pm 0.012$ | $0.588 \pm 0.033$ | |
| | Temporal $BALC_{KLD}$ | $0.511 \pm 0.030$ | $0.496 \pm 0.023$ | $0.496 \pm 0.023$ | $0.503 \pm 0.010$ | $0.532 \pm 0.027$ | |
| $\mathcal{D}_5$ | $BALD_{MCD}$ | $0.891 \pm 0.003$ | $0.893 \pm 0.002$ | $0.901 \pm 0.001$ | $0.899 \pm 0.003$ | $0.902 \pm 0.003$ | $0.898 \pm 0.002$ |
| | $BALD_{MCP}$ | $0.895 \pm 0.002$ | $0.894 \pm 0.003$ | $0.904 \pm 0.003$ | $0.897 \pm 0.003$ | $0.900 \pm 0.002$ | |
| | $BALC_{KLD}$ | $0.894 \pm 0.003$ | $0.895 \pm 0.001$ | $0.899 \pm 0.002$ | $0.895 \pm 0.001$ | $0.899 \pm 0.002$ | |
| | Temporal $BALC_{KLD}$ | $0.895 \pm 0.001$ | $0.894 \pm 0.002$ | $0.899 \pm 0.002$ | $0.896 \pm 0.002$ | $0.899 \pm 0.004$ | |

## 6.3 ACTIVE LEARNING WITH NOISY ORACLE

So far, we have presented scenarios in which oracles are either absent or present with the ability to provide accurate labels. In healthcare, however, physicians may be ill-trained, fatigued, or unable to diagnose a case due to its difficulty. We simulate this scenario by introducing two types of label noise. We stochastically flip each label to 1) any other label randomly (**Random**), or 2) its nearest neighbour from a *different* class in a compressed subspace (**Nearest Neighbour**). Whereas the first form of noise is extreme, the latter is more realistic as it may represent uncertain physician diagnoses. To simulate various magnitudes of noise, we chose the probability of introducing noise, $\gamma = [0.05, 0.1, 0.2, 0.4, 0.8]$. In Fig. 6, we illustrate the effect of label noise on the test AUC for the various oracle questioning strategies.

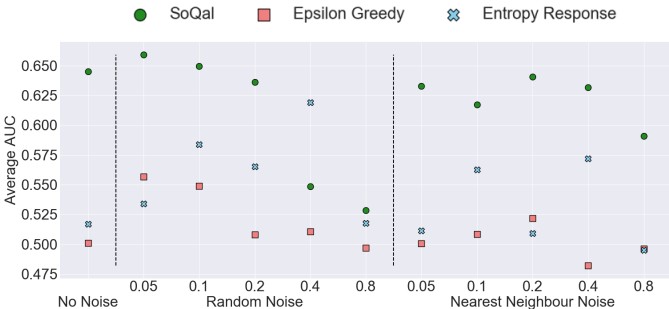

Figure 6: Average test AUC for the oracle questioning strategies in the absence and presence, of various magnitudes, of label noise on $\mathcal{D}_1$ using $\text{BALD}_{\text{MCP}}$. We show that with up to $80\%$ random or nearest neighbour label noise, SoQal still outperforms the remaining methods when trained *without* label noise. This illustrates that SoQal is better equipped to deal with oracle label noise.

We find that SoQal outperforms the remaining strategies regardless of noise type and magnitude (except at $40\%$ random noise). For example, at $5\%$ random noise, SoQal achieves an AUC $\approx 0.66$ whereas Epsilon Greedy and Entropy Response achieve an AUC $\approx 0.56$ and $\approx 0.53$, respectively. Surprisingly, we find that the introduction of label noise can sometimes improve performance. For example, SoQal's AUC $\approx 0.64 \rightarrow 0.66$ with no noise and $5\%$ random noise, respectively. We hypothesize that this is due to inherent label noise in the datasets. By introducing further noise, we nudge these labels towards their ground-truth values. Moreover, SoQal is better able to deal with label noise than its counterparts. Specifically, SoQal at $80\%$ random noise achieves AUC $\approx 0.53$ whereas Epsilon Greedy and Entropy Response trained *without noise* achieve AUC $\approx 0.50$ and $\approx 0.52$, respectively. This effect, which is even more pronounced when dealing with nearest neighbour noise, indicates the utility of SoQal in the presence of a noisy oracle. We arrive at similar conclusions when experimenting with other datasets and acquisition functions (see Appendix O).

## 7 DISCUSSION AND FUTURE WORK

In this paper, we proposed a novel consistency-based active learning framework which perturbs both inputs and network parameters and acquires instances to which the network is least robust. We illustrate the utility of this approach in the *absence* of an oracle. Moreover, we propose a strategy that dynamically determines whether an oracle should be requested for a label. We empirically show that this approach is better able to deal with noisy oracles than the baseline methods. We now elucidate several future avenues worth exploring.

**Incorporating Prior Information.** The default mode for SoQal is deferral to an oracle. However, relevant a priori information, such as the degree of noise inherent in the oracle's labels, can be incorporated to alter either the default mode or the Hellinger threshold, $S$.

**Incorporating Multiple Oracles.** In this work, we focused on the presence of a single oracle. Scenarios in which multiple oracles exist may better reflect clinical environments which include multiple experts of various levels of competency. Therefore, dynamically querying these oracles might be of interest.

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
