# OpenReview forum: "SoCal: Selective Oracle Questioning for Consistency-based Active Learning of Cardiac Signals"
_ICLR.cc/2021/Conference — Reject_

### Official Review · AnonReviewer3 · 2020-10-25
**Better to see more types of datasets and network architectures, and how about pseudo labels augmentation?**

**Rating:** 4
**Confidence:** 4

**Review:**

The authors proposed a consistent-based active learning framework to annotate largely unlabeled physiological signals with the help of human annotators (oracles). The paper is well organized and easy to follow. It is somewhat novel to equipping active learning with consistency learning and selective classification. The experiments (along with the Appendix) give a comprehensive analysis of the method.

However, I do have the following concerns:

First, there might be a simple baseline to deal with unavailable labels: to augment all unavailable data with pseudo labels (such as predictions of the current model, or assign with nearest neighbor’s label), and train on [oracle labeled data] + [pseudo labeled data] together. How is this method compared with yours?

Second, both title and abstract emphasize physiological signals. But the proposed method is not related to physiological signals. And it seems no special design for physiological signals. Only all four experimental datasets are physiological signals - ECG and PPG. However, if it is targeting physiological signals, others like blood pressure (BP), heart rate (HR), and electroencephalogram (EEG) are also important physiological signals. It is necessary to see how this method performs on those data.

Third, datasets have different sampling rates, but this paper used the same model architecture (in Appendix H.1). And such architecture is not a first-choice model for any of the four datasets. For D1, D2, D3, one can find many other delicate designed models from the challenge papers. For D4, Hannun et al., 2019 also used a much deeper neural network. Thus, in Table 1, we can see that even 100% Oracle is much lower than reported numbers in other papers.

Some minors from Table 1: For AUC, do you mean area under ROC-curve, precision-recall curve, or the others? If it is ROC-AUC, it is confusing that table 1 AUCs of most methods on D4 are lower than 0.5 - even worse than a random classifier. Besides, BALC seems no obvious positive effect than BALD.

---

> ### Author Response · Authors · 2020-11-18
> **Response to Reviewer 3 - Round 1**
>
> We thank the reviewer for taking the time and effort to review the manuscript and for providing us with valuable feedback. We address your comments below.
>
> **SIMPLE BASELINE**
> Our experiments in the **absence** of an oracle implement what you have suggested in a dynamic manner. More specifically, it assigns pseudo-labels to the unlabelled instances using the existing network’s parameters. So we end up training a network that is solely based on [oracle labelled data] + [pseudo-labelled data] as you suggested. However, assigning pseudo-labels to all unlabelled instances and incorporating those into the training set would be an unfair comparison to the existing methods in the paper. This is primarily because of the discrepancy that would arise due to the number of instances that are used for training.
>
> **OTHER DATASETS**
> We modify the manuscript to illustrate our explicit focus on cardiac signals, which can be thought of as a subset of physiological signals. These include the PPG and ECG modalities. To address your concern about evaluating on different datasets, we conduct our experiments on CIFAR10, as was recommended by other reviewers. These results can be found in Sec. 6.1 and 6.2.
>
> **NETWORK ARCHITECTURE**
> It is important to note that comparisons in AUC cannot be made across different papers when it comes to several of these datasets. This is primarily due to data availability. For example, the Hannun et al. 2019 paper only released their test set to the public. This means we can only use their test set for (training, validation, and testing in our scenario). Moreover, some PhysioNet challenges do not release their final test set to the public even after the challenge is completed. Therefore, reported AUC values cannot be compared. As for the AUC values that we report, they are indeed AUROC values. As for their dip below 0.5, that is technically possible. If, for example, an AUROC = 0.40 in a binary prediction task, that simply implies that if all of a network's predictions were flipped, then an AUROC = 1 - 0.40 = 0.60, would be achieved.
>
> We hope the above responses and the modified version of the manuscript have addressed your concerns.

---

### Official Review · AnonReviewer4 · 2020-10-27
**paper is not self-contained, important technical details are missing, novel contributions are not clear**

**Rating:** 4
**Confidence:** 3

**Review:**

This paper proposes the framework that dynamically determines whether to request a label from an oracle or to generate a
pseudo-label instead. It is claimed to remain strong performance with noisy oracle. In this work, the unlabelled instances in proximity to the decision boundary are considered more informative. After performing Monte Carlo Perturbations on unlabeled data and feeding into networks, the samples with significant different outputs are considered close to the boundary.  This paper proposes to perturb both instances and parameters as shown in Equation (1)(2).

Important technical details are missing. For example, based on the two Gaussian distributions of perturbed networks and the Hellinger distance of the distributions to determine the probability of requesting a label from an oracle as in Equation (5). It is not mentioned how to generate the pseudo-label. The connection of the probability in Equation (5) and previous sections are not close.

My main concern is that the novel contribution of this work is not strong. Though this work proposed to consider both the instance perturbation, parameters perturbation, temporal information and so on. They are not strongly motivated and no related references to support the motivations. There are also no related empirical experiments to support the motivations.

---

> ### Author Response · Authors · 2020-11-18
> **Response to Reviewer 4 - Round 1**
>
> We thank the reviewer for taking the time and effort to review the manuscript and for providing us with valuable feedback. We address your comments below.
>
> **NOVELTY**
> We believe our paper offers several contributions. First, we propose to apply perturbations to both inputs and network parameters simultaneously and provide an in-depth discussion of why this can be beneficial in the modified version of the manuscript (Sec. 4.1.2). Please also refer to Fig. 2 in Sec. 4.1.2 to complement the explanation of the motivation underlying our proposed methods (BALC, in particular). Second, we leverage these perturbations to design a family of divergence-based acquisition functions (BALC). These acquisition functions acquire instances to which the network is least robust. Finally, we design a dynamic oracle selection strategy that sits atop both existing acquisition functions and those we have introduced in order to reduce the labelling burden placed on oracles (e.g., physicians). Overall, we illustrate the potential of our methods on multiple cardiac time-series datasets.
>
> **TECHNICAL DETAILS**
> Can the reviewer please be more specific about which components of the methods section are unclear? We are happy to clarify anything that may be unclear.
>
> If your question pertains to the oracle selection component in particular, we address that here. The crux of this approach lies in the module entitled oracle selection network, h, shown in Fig. 3. This module outputs a scalar value, t, for each instance in the dataset. In the presence of labelled data, this module is trained by 'predicting the zero-one loss' of the prediction network, g. In doing so, high values of the output of the oracle selection network (i.e., close to 1) tend to be associated with instances that are incorrectly classified by the network. Therefore, this output can act as a proxy for **unlabelled** instances that the network is unable to classify and thus would intuitively benefit from an oracle label. However, how would we go about setting a threshold on this output/proxy in order to select an oracle? Naively, you could say that if the output is greater than 0.5, then you should always select an oracle. Such a naive threshold, however, may not be appropriate especially if the output of the oracle selection network is biased towards the extremes (i.e., 0 or 1). In such an event, all unlabelled instances would either be labelled by an oracle or pseudo-labelled by the network, and in turn we would lose the benefit of a dynamic selection system.
>
> To avoid this bias and ensure the reliability of the oracle selection network outputs as a proxy, we do the following: A) we fit a Gaussian to the distribution of these outputs in response to **labelled** instances (Fig. 4b). If these distributions are separable, that implies that we have a reliable proxy (one that can distinguish instances that can be correctly classified by the network and those that cannot). B) We quantify the separability of these distributions by using the Hellinger distance (Fig. 4c). Since high separability equates to high proxy reliability, we depend on the proxy only when separability (Hellinger distance) is high enough (i.e., Dh $\geq$ S). C) Dependence on the proxy means we allow the network to make the decision of whether to ask for an oracle label or to pseudo-label instead. To determine which of these two decision is made, we follow Eq. 5 (centre line). By evaluating the two Gaussian distributions at the output, t, of the oracle selection network for each **unlabelled** instance, and comparing their values, we can make the oracle selection decision. This entire explanation can be found in the subsection entitled 'Decision-making with proxy' in Sec. 4.3.
>
> We hope the above responses and the modified version of the manuscript have addressed your concerns.

---

### Official Review · AnonReviewer1 · 2020-10-28
**Interesting idea but the empirical evidence is inconclusive**

**Rating:** 5
**Confidence:** 4

**Review:**

The paper proposes an active learning framework which relies on a consistency-based acquisition function and a selective querying method. The idea is interesting and the paper is well written; however, I have the following comments and questions:

he motivation and intuition behind the proposed acquisition functions should be in the main text instead of appendix as this is one of the main contributions of the work (at least partially). Also Figure 6 in the Appendix does not really provide an intuition; it’s basically showing the description in the text by the figure. The terms NSR, AFib and LBBB are introduced in the figure without any definition in the text (I believe these are your e cardiac arrhythmia labels but the reader not familiar with these terms may have trouble figuring it out.). For a more intuitive figure, I suggest using a synthetic toy example.

I believe the ‘selective query questioning’ module is quite similar to the approach proposed by Yoo and Kweon in ‘Learning Loss for Active Learning’ in CVPR 2019. The main difference is that the loss prediction module in the latter is more general and can support tasks other than classification. Can you elaborate on the advantages of your approach over theirs?

Can you also provide the same analysis that you’ve done for D2 and D1 in Fig 4(a, b) for other datasets and \beta values? I couldn’t find it in the appendix.  The results in the Appendix (I) do not provide a coherent story which makes it hard to judge the pros and cons of the method. Also it would be nice if you add some more standard datasets such as CIFAR-10 as it helps comparing the method with many other active learning baselines in the literature.

It’s a bit unintuitive to think about examples that the acquisition function is considering informative but the ‘selective oracle questioning’ is preferring pseudo labels for them over oracle labels. In other words, why should these two modules disagree on an instance? If we are highly uncertain about the label of an instance under what scenarios the pseudo label for that instance helps the model? What’s the advantage of your approach over using an approach which only has an acquisition function based on loss prediction (e.g. ‘Learning Loss for Active Learning’ )?

Experiments in 6.2: Do you observe the same behavior for other beta thresholds?

Overall, I think the paper is generally well written and the idea is promising but I believe the strengths and weaknesses of the method need to be discussed in more details. It’s totally fine that the model fails on some datasets and shows significant improvement over baselines in some other datasets; however, in the current version of the paper the reader is not able to understand what are the success and failure chances of the method. The experiments in the appendix fail to provide a coherent story or a discussion on why the method is not performing well in some experiments (e.g. ~40% of the experiments in test set performance without an oracle).

-----------------------------------------------
Post-rebuttal comments:
I'd like to thank the authors for adding the experiments; the paper looks stronger now but unfortunately, the results on the new experiments are not that encouraging. Considering the results and also other reviews; I'd like to keep my score as marginally below acceptance threshold.

---

> ### Author Response · Authors · 2020-11-18
> **Response to Reviewer 1 - Round 1 (Part 1)**
>
> We thank the reviewer for taking the time and effort to review the manuscript and for providing us with valuable feedback. We address your comments below.
>
> **MOTIVATION AND JUSTIFICATION OF METHODS**
> We have provided an in-depth description of the motivation underlying our methods (Sec. 4.1.2). As for Fig. 6 (that was previously in Appendix D), we have now included that in the main manuscript and modified it to complement the explanation of the motivation of our methods (BALC, in particular). We have also replaced any medical jargon (e.g., NSR, Afib, etc.) in that figure that was present before with more generic labels (e.g., A, B, and C) to assist understanding for readers in the ML community.
>
> **SIMILARITY OF OUR METHOD TO 'LEARNING LOSS' PAPER**
> We thank the reviewer for bringing this paper to our attention. Indeed, their loss prediction module is quite similar to our oracle selection network in terms of design. Our experimentation with classification tasks explains our use of the zero-one loss as the target loss of choice. However, our method would still be able to accommodate other tasks (e.g., regression) by using a mean-squared error loss as the target loss. In this scenario, one can continue to use Eq. 5 as per normal to calculate the probability of asking an oracle to label an instance. This is because the Hellinger distance maps to [0-1].
>
> We also believe the method introduced in ‘Learning Loss for Active Learning’ is limited in several ways. To begin, the reliability of their loss-prediction module as an acquisition function is arguable. For example, at the beginning of neural network training/when few datapoints are used, the loss prediction module has yet to be trained sufficiently and thus is likely to perform poorly on all unlabelled instances. This makes it difficult to distinguish between unlabelled instances solely based on their relative predicted loss values. The same argument can be applied toward the end of neural network training/when many datapoints are used. The network has become adept at solving the task at hand and thus predicts low loss values for the unlabelled instances. Once again, this makes it difficult to distinguish between unlabelled instances when it comes to acquiring them for annotation. In contrast, our method is not dependent upon the potentially unreliable loss prediction to choose which instances to acquire, an integral decision in the active learning framework. In fact, we separate the process of instance selection and oracle selection entirely. We decide to acquire instances using a consistency-based acquisition function and select an oracle to label based on the oracle selection network. To avoid an unreliable oracle selection network, we are constantly evaluating its reliability (as exemplified by Fig. 4b and Eq. 5) and making a decision based on that. On a related note, the method introduced in ‘Learning Loss for Active Learning’  always asks for a label from an oracle. In addition to being burdensome, this request may not always be appropriate, for example, in the case of a noisy oracle or an absent one. In contrast, we provide the network with the option of abstaining and we show that this act of abstention can be beneficial (e.g., see results in Fig. 6).
>
> **FURTHER ANALYSIS**
> We have included the remaining validation AUC curves for all datasets and all beta values in Appendix J. As for more standard datasets, we have taken your advice and implemented our family of methods on the CIFAR10 dataset both with and without an oracle. These results can be found in Sec. 6.1 and 6.2.
>
> **INTUITION ON MODULES**
> Roughly speaking, your initial question on the disagreement of modules is based on the assumption that informative instances are likely to have a high loss, and those with a high loss will likely necessitate an oracle label. This is not necessarily the case. When acquiring instances, it is the relative value of the acquisition function across all unlabelled instances that matters. Therefore, an instance can be ranked as highly informative by an acquisition function yet still be associated with a low loss value, and thus potentially obviating the need for an oracle to label that instance. To illustrate this point, take the following example: we have instances X and Y on the same side of the decision boundary. Let us assume X is closer (but not too close) to the decision boundary than Y is. As a result, X is deemed more informative than Y and thus is acquired. However, since X is far enough from the decision boundary, it can still be associated with a low loss value and thus obviate the need for an oracle to label it. With our method, this scenario is more likely to occur as neural network training progresses and the network becomes more adept at solving the task at hand/classifying instances.
>
> Response is continued in Part 2.

---

> > ### Author Response · Authors · 2020-11-18
> > **Response to Reviewer 1 - Round 1 (Part 2)**
> >
> > **EXPERIMENT IN SECTION 6.2**
> > We initially chose to conduct these particular experiments at beta=0.1 for several reasons. First, the presence of a noise-free oracle meant that we were less likely to encounter the cold-start problem. This is because the network was guaranteed to obtain accurate labels to be used for training purposes. Second, this scenario illustrates the relatively most extreme situation that a network can find itself in (low data regime), and thus the method that prevails here is the one that is most robust to data scarcity.
> >
> > We hope the above responses and the modified version of the manuscript have addressed your concerns.

---

### Official Review · AnonReviewer2 · 2020-10-28
**Incremental in Novelty, Good Empirical Performance**

**Rating:** 5
**Confidence:** 3

**Review:**

The paper proposes an active learning framework called SoCal that is consistency-based and can decide between whether to make use of the oracle to provide a label or to make use of a pseudo-label generated by the algorithm itself instead. The proposed method hopes to address resource-constrained active learning scenarios where the oracle is not always available or we wish to make use of the oracle as infrequently as possible. Experimental results demonstrate reasonable performance on four publically available physiological datasets.  Experimental results when the oracle is noisy is also reported.

Overall, I  think the paper proposes a reasonable approach to tackle various challenges one might encounter in the resource-constrained active learning setting. I have the following concerns:

1. Perhaps the major concern I have is the novelty of this paper. While the paper proposes a framework that harnesses several elements in design that are helpful to its success, most of these elements are not new to the literature. Perturbation in samples and network parameters are widely used practices in achieving consistency. Deciding to query the oracle or not is also a problem studied in active learning.

2. Related to the first point, because of the lack of obvious novelty in the paper, providing insights and demonstrating why the proposed composition of design elements should work is all the more important. However, I find that the paper is not very insightful. There is a lack of justification and insights in explaining why the proposed method should work. As a remedy, theoretical justification of some sorts and ablation studies will be desirable.

Miscellaneous:
1. Why the Gaussian assumption made in the paper is reasonable?
2. Any thoughts on why MCP or MCD alone "can fail to identify instances that lie in the region of uncertainty"?
3. When there is no oracle, or when there is a noisy oracle, is it reasonable to compare the proposed method with methods that deal with noisy labels such as semi-supervised learning?
4. While I understand that learning from physiological signals can be an important use case for the proposed method, the proposed method also seems to be general enough to be applied to other non-healthcare datasets. Is there any consideration on why or why not running the proposed methods on these datasets?


=======Post Discussion==========

After reading the author's response and other reviewers' reviews, I still find the novelty of this paper somewhat insufficient. Therefore, I would like to maintain my initial evaluation.

---

> ### Author Response · Authors · 2020-11-18
> **Response to Reviewer 2 - Round 1**
>
> We thank the reviewer for taking the time and effort to review the manuscript and for providing us with valuable feedback. We address your comments below.
>
> **NOVELTY**
> The concept of applying perturbations to inputs or network parameters alone has been introduced before in the literature. Our work extends beyond the application of such perturbations in three distinct ways. First, we propose to apply perturbations to both inputs and network parameters simultaneously. We provide an in-depth discussion of why this can be beneficial in the modified version of the manuscript (Sec. 4.1.2). Second, we leverage these perturbations to design a family of divergence-based acquisition functions (BALC). These acquisition functions acquire instances to which the network is least robust. Finally, we design a dynamic oracle selection strategy that sits atop both existing acquisition functions and those we have introduced in order to reduce the labelling burden placed on oracles (e.g., physicians). We do not dispute the fact that querying an oracle in and of itself has been studied in active learning. We simply design a flexible strategy for doing so and illustrate its potential on multiple cardiac time-series datasets.
>
> **MOTIVATION AND JUSTIFICATION OF METHODS**
> We provide an in-depth description of the motivation underlying the proposed methods in the modified version of the manuscript (Sec. 4.1.2). Moreover, we hope Fig. 2 and its associated explanation can help the reviewer better understand the limitations of existing approaches (e.g., Monte Carlo Dropout) and elucidate the potential approaches that perturb both inputs and network parameters.
>
> **GAUSSIAN ASSUMPTION**
> After having conducted several experiments, we empirically observed that the distribution of outputs from the oracle selection network, g, could be reasonably approximated by a Gaussian distribution. This is supported, for example, by the two distributions shown in Fig. 3b. We also found that this Gaussian assumption is more likely to hold when the training set was quite large, as would be expected.
>
> **REGION OF UNCERTAINTY**
> As for why MCP and MCD can "fail to identify instances that lie in the region of uncertainty", we illustrate this point with the following example: without loss of generality, let us assume an unlabelled instance is in proximity to some decision boundary, A, and is classified by the network as belonging to some arbitrary class 3. Such proximity should deem the instance informative for the training process. In the MCD setting, perturbations are applied to parameters, generating various decision boundaries, which in turn influence the network outputs. In Fig. 2 (red rectangle), we visualize such outputs for three arbitrary classes. If these parameter perturbations happen to be too small in magnitude, for example, then the network will continue to classify the instance as belonging to the same class. At this stage, regardless of whether an uncertainty-based or a consistency-based acquisition function is used, the instance would be deemed uninformative, and thus not acquired. As a result, an instance that should have been acquired (due to its proximity to the decision boundary) was erroneously deemed uninformative. A similar argument can be extended to MCP.
>
> We include this explanation in the modified version of the manuscript (Sec. 4.1.2)
>
> **SEMI-SUPERVISED LEARNING**
> Although active learning does technically fall under the auspices of semi-supervised learning, we believe our method is distinct from the latter for the following reason. In the noisy oracle scenario, our selective oracle questioning method is quite flexible as it allows for the network to dynamically adapt to labels being provided by the oracle. Static semi-supervised approaches can be thought of as a special case of our method where one would pre-train once on the unlabelled data, fine-tune on the labelled data, then pseudo-label the unlabelled data.
>
> **HEALTHCARE DATASETS**
> The focus of the paper is indeed on cardiac time-series signals (PPG and ECG). Such a focus was chosen due to the large amount of unlabelled data that exist within the medical domain. Having said that, we implemented our methods on the CIFAR10 dataset in both scenarios with and without an oracle and present these results in the modified version of the manuscript (Sec. 6.1 and 6.2). In Sec. 6.1, we find that BALC_KLD continues to perform well in this setting and that the incorporation of temporal information can benefit the performance of static acquisition functions. In Sec. 6. 2, we find that our selective oracle questioning method is on par with the remaining methods on the CIFAR10 dataset.
>
> We hope the above responses and the modified version of the manuscript has addressed your concerns.

---

### Decision · Program_Chairs · 2021-01-07
**Final Decision**

**Decision:**

Reject

**Comment:**

The reviewers still have several concerns about the paper after the author feedback stage: the novelty of the paper is not sufficient; the experimental results are not very encouraging. We encourage the authors fixing these issues in the next revision.